# Network Pharmacology and Molecular Docking-Based Approach to Explore Potential Bioactive Compounds from *Kaempferia parviflora* on Chemokine Signaling Pathways in the Treatment of Psoriasis Disease

**DOI:** 10.3390/ijms26115243

**Published:** 2025-05-29

**Authors:** Chotiwit Sakuludomkan, Jittasak Khowsathit, Pilaiporn Thippraphan, Nut Koonrungsesomboon, Mingkwan Na Takuathung, Weerakit Taychaworaditsakul

**Affiliations:** 1Clinical Research Center for Food and Herbal Product Trials and Development (CR-FAH), Faculty of Medicine, Chiang Mai University, Chiang Mai 50200, Thailand; chotiwit.cs@gmail.com (C.S.); nut.koonrung@cmu.ac.th (N.K.); mingkwan.n@cmu.ac.th (M.N.T.); 2Department of Pharmacology, Faculty of Medicine, Chiang Mai University, Chiang Mai 50200, Thailand; 3Department of Biochemistry, Faculty of Medicine, Chiang Mai University, Chiang Mai 50200, Thailand; jittasak.khowsathit@cmu.ac.th (J.K.); tipprapant@gmail.com (P.T.)

**Keywords:** *Kaempferia parviflora*, network pharmacology, molecular docking, inflammation, chemokine signaling, psoriasis

## Abstract

Psoriasis is a chronic inflammatory skin disorder characterized by keratinocyte hyperproliferation and dysregulated chemokine signaling. *Kaempferia parviflora* (KP) has long been valued for its medicinal properties; however, its specific role in psoriasis treatment remains unclear. This study investigates the anti-psoriatic potential of methoxyflavones derived from KP through an integrated approach combining network pharmacology, molecular docking, and experimental validation. A total of 232 target genes were identified as being associated with KP bioactive compounds, of which 64 overlapped with psoriasis-related genes implicated in chemokine signaling pathways. Molecular docking analyses revealed that key methoxyflavones interact with pivotal proteins such as protein kinase B (AKT1 or AKT), proto-oncogene tyrosine-protein kinase (SRC), and phosphoinositide-3-kinase regulatory subunit 1 (PIK3R1), suggesting their potential involvement in modulating inflammation. Experimental results confirmed that 5,7,4′-trimethoxyflavone and 3,5,7-trimethoxyflavone significantly inhibited keratinocyte proliferation, migration, and macrophage activation, key processes in psoriasis progression. Additionally, both compounds reduced nitric oxide production, supporting their anti-inflammatory effects. Western blot analysis further demonstrated that these compounds tended to decrease the phosphorylation levels of AKT and SRC, supporting their role in influencing inflammatory signaling pathways. These findings suggest that methoxyflavones from KP act through multi-target mechanisms, offering potential as natural therapeutic agents for psoriasis. Further, in vivo studies are needed to validate their efficacy and explore their clinical applications.

## 1. Introduction

Psoriasis is a chronic inflammatory skin disease with a prevalence of up to 11.43% in some regions of the world, making it a serious global concern impacting at least 100 million people worldwide [1,2]. Studies indicate that psoriasis incidence is influenced by aberrant expression of psoriasis-susceptible genes, autoimmune dysregulation, obesity, and abnormalities in various inflammatory signaling pathways [3]. Despite controversies regarding the pathophysiology of psoriasis, it has been demonstrated that an aberrant immune response to specific skin factors leads skin cells to divide more rapidly than usual, resulting in a thickening of the rash [4]. The role of chemokines in psoriasis inflammation is essential, as they regulate both immune cell migration and activation. Importantly, chemokines are pivotal in psoriasis-related inflammation by regulating immune cell migration and activation. These chemokines facilitate the recruitment and activation of T cells, macrophages, and neutrophils in the skin through pathways such as extracellular signal-regulated kinase (ERK) for keratinocyte proliferation and phosphoinositide 3-kinase (PI3K)/AKT/mechanistic target of rapamycin (mTOR) for immune cell recruitment [5,6,7,8]. In addition, interleukin (IL)-17 produced by T helper (TH)-17 cells binds to its receptor, interleukin 17 receptor (IL-17R), which is abundantly expressed in keratinocytes, thereby triggering key transcription factors such as PI3K/AKT, nuclear factor kappa B (NF-κB), and cellular Jun proto-oncogene (c-Jun). Activated keratinocytes subsequently release inflammatory cytokines (e.g., IL-6, IL-1β, and tumor necrosis factor alpha (TNF-α)) and chemokines (e.g., C–X–C motif chemokine ligand (CXCL)-1, CXCL2, C–C motif chemokine ligand 20 (CCL20), and CXCL8), which further interact with immune cells to establish and sustain the immunological circuits of psoriasis [9]. Given the biological significance of immune signaling in disease pathogenesis, targeting this pathway offers a promising therapeutic strategy for drug development. Notably, chemokines, their receptors, and associated signaling molecules have been identified as potential therapeutic targets [10].

In accordance with the pathophysiology and treatments of psoriasis, anti-hyperproliferative agents are among the most commonly used treatments for severe plaque psoriasis, including methotrexate, tacrolimus, and cyclosporine A, along with phototherapy [11,12,13]. However, prolonged use is associated with severe side effects, disease recurrence, and cost-related challenges [14]. Therefore, the current medical challenge is to find a strategy to proactively treat psoriasis. In recent years, significant progress has been made in the understanding of the therapies for inflammation-related psoriasis, particularly through network pharmacology investigations.

Network pharmacology, which attempts to comprehend drug actions and interactions with multiple targets, has recently evolved [15,16]. The mathematical framework of the network is well established and may be utilized to comprehend the topological aspects of the interconnected biological system, such as diseases, genes, proteins, and molecules. This framework led to globally relevant findings and trends beneficial for drug discovery [17,18,19,20]. In modern medicine, the application of network pharmacology to herbal medicine creates a new paradigm for understanding and elucidating the underlying complex relationships between botanical formulas and their mechanisms of action and between bioactive substances and associated targets [21,22]. Apart from the network pharmacology approach, molecular docking is utilized for the prediction of therapeutic agents to their binding sites on disease-related molecular targets; this approach yields insights into the structure–activity relationship (SAR) as well as binding kinetics [23]. Thus, network pharmacology and molecular docking may be applied together to identify potential molecular targets on signaling pathways.

*Kaempferia parviflora* (KP) or Krachaidam, also known as black ginger, which belongs to the Zingiberaceae family, is native to the north and northeast of Thailand [24]. According to ethnopharmacological reports, KP rhizome extracts have been shown to have a variety of pharmacological activities, including adipocyte hypertrophy suppression [25], anti-allergic [26], and anti-inflammatory properties [27]. These pharmacological effects of KP are attributed to the presence of numerous methoxyflavone derivatives [28]. For example, methoxyflavones inhibited the generation of nitric oxide (NO), inducible nitric oxide synthase (iNOS), prostaglandin E_2_ (PGE_2_), and TNF-α via activation of the spleen tyrosine kinase (SYK) pathway but not the ERK or c-Jun N-terminal kinase (JNK) pathways in lipopolysaccharide (LPS)-induced RAW264.7 cells [29]. Although many studies on KP extract have demonstrated its anti-inflammatory properties, there is limited evidence on the molecular mechanisms of its active compounds in regulating psoriasis-related chemokine signaling pathways. Therefore, the discovery of potentially therapeutic molecular targets from KP for psoriasis-related inflammatory illnesses presents an interesting and promising area of research.

To this end, combining network pharmacology in combination with molecular docking to investigate the anti-inflammatory action of potential KP compounds on psoriasis would be of great value. Our objective is to explore these potential compounds from KP using integrated network pharmacology and molecular docking analysis to determine their probable molecular mechanism of action as treatments or preventive strategies for psoriasis. The findings of this study may lead to the development of alternative psoriasis treatments based on herbal medicine.

## 2. Results

### 2.1. Bioactive Components, Target Prediction of KP, and Screening of Psoriasis-Related Chemokine Target Genes

A total of 16 methoxyflavones in KP were obtained through literature searches [27,28,30], and the relevant ADME information is presented in Table 1. All methoxyflavones met the criteria with Lipinski’s Rules of Five (molecular weight < 500 g/mol, HBA < 10, HBD < 5, MlogP < 4.15, and TPSA (Å2) < 140). Furthermore, these compounds demonstrated a high bioavailability score (>0.1) and no Lipinski violations. Predicting potential pharmacological adverse effects is a crucial step in the drug development process. Thus, the ADMETlab 2.0 software was used for evaluating the toxicological parameters of these compounds, such as carcinogenicity, mutagenicity, hepatotoxicity, and rat oral acute toxicity. All the methoxyflavones have no carcinogenicity, hepatotoxicity, or rat oral acute toxicity. However, only 5,7-dimethoxyflavone had medium mutagenicity (Appendix A). According to the target prediction systems in SwissTargetPrediction, STITCH, and TargetNet, we found 232 genes that could be the target of the 16 methoxyflavones (Appendix A). The KP component–target network was constructed by Cytoscape 3.10.1 software, including 249 nodes and 2207 edges. The black circular node represents the active compounds, and the blue circular node represents the target genes (Figure 1A). Among them, topological analysis by a network analyzer showed that the top five active compounds in this network are 5,7,4′-trimethoxyflavone (degree = 157), 3,5,7,3′,4′-pentamethoxyflavone (degree = 156), 5-hydroxy-7,3′,4′-trimethoxyflavone (degree = 155), 5-hydroxy-3,7,4′-trimethoxyflavone (degree = 154), and 5-hydroxy-7-methoxyflavone (degree = 152). This finding suggests that the main bioactive compounds in KP were essential to the treatment of related disorders through a synergistic action.

Psoriasis-related chemokine target genes were screened from human genomic databases, and 1084 genes at the intersection of these databases were obtained. In this protein–protein interaction (PPI) network analysis, there were 1058 nodes and 30,937 connection edges. The increasing number of edge responses to a higher degree, in which core genes (black) have higher degrees (Figure 1B). Among them, the top ten highest degrees connected in the network were *TNF* (degree = 457), *AKT1* (degree = 436), *IL6* (degree = 426), *IL1-beta* (degree = 375), *CD4* (degree = 369), *SRC* (degree = 342), *EGFR* (degree = 335), *VEGFA* (degree = 325), *STAT3* (degree = 324), and *TP53* (degree = 324), indicating that these targets are significant contributors to the pathogenesis of psoriasis.

### 2.2. PPI Network of Target Genes

The intersection between 232 KP targets and 1084 psoriasis-related chemokine genes was analyzed using the Venn diagram tool [31]. As shown in Figure 2A, there were 64 overlapping targets considered core targets associated with both KP bioactive compounds and psoriasis-related chemokines. Analysis of protein–protein interactions among the psoriasis-related chemokine targets was performed using a string database, and the result is shown in Figure 2B. The black-to-blue nodes in the middle represent the top 10 genes with the largest degree values (i.e., *TNF*, *AKT1*, *SRC*, *CASP3*, *ESR1*, *MMP9*, *PPARG*, *PIK3R1*, *CDC42*, and *RELA)*, which play a key role in the core target of KP (Figure 2C). The KEGG pathway, in addition to the core target PPI of KP, suggested that chemokine signaling pathways were directly related to psoriasis disease. PI3K-AKT1 is involved in the regulation of multiple cellular physiological processes by activating downstream corresponding effector molecules. This chemokine signaling target pathway was therefore selected for further analysis. As shown in Figure 2D, the PPI network analysis of 13 core targets in the chemokine signaling pathway identified based on the degree value from high to low were *AKT1* (degree = 11), *SRC* (degree = 9), *CDC43* (degree = 8), *PIK3R1* (degree = 8), *GSK3B* (degree = 7), *PTK2* (degree = 7), *RELA* (degree = 5), *PIK3CG* (degree = 5), *RAC1* (degree = 5), *IKBKB* (degree = 4), *GSK3A* (degree = 3), *CXCR1* (degree = 2), and *GRK6* (degree = 2) (Appendix A).

### 2.3. Gene Ontology (GO) and KEGG Pathway Enrichment Analyses

To elucidate the biological functions of KP target genes involved in psoriasis-related chemokine targets, the DAVID database was used to perform GO analysis. This analysis yielded a total of 409 GO results, which included 296 entries for biological process (BP), 43 entries for cellular component (CC), and 70 entries for molecular function (MF). The top 15 terms from these GO categories are presented in Figure 3A and Appendix A. The biological processes (BP) linked to KP primarily pertain to inflammatory response, while the cellular components of KP are predominantly associated with the plasma membrane. The molecular functions chiefly involve transmembrane receptor protein tyrosine kinase activity. Additionally, KEGG pathway enrichment analysis was conducted on the 64 target genes, and the top 20 pathways were visualized using Cytoscape 3.9.0, represented as an octagon plot, as shown in Figure 3B. The analysis of these top 20 KEGG pathways revealed that the most enriched pathway was associated with psoriasis-related chemokine targets, including the chemokine signaling pathways and the PI3K-AKT signaling pathways (Appendix A). The identified targets, shown in red, were implicated in two major pathological processes related to psoriasis. Other significant pathways included the T and B cell receptor signaling pathway, TNF-α signaling pathway, IL-17 signaling pathway, NF-κB signaling pathway, Th17, Th1, and Th2 cell differentiation, and Janus kinase (JAK)/signal transducer and activator of transcription (STAT) signaling pathways (Appendix A).

### 2.4. In Silico Analysis of Bioactive Compounds in KP on Psoriasis-Related Chemokine Target Genes 

Molecular docking and binding energy prediction were performed with bioactive compounds and representative targets to validate the findings from network pharmacology results. Based on the high degrees of core targets in the KEGG results PPI network analysis, promising targets such as AKT1, SRC, and PIK3R1 were selected for molecular docking. Our results indicate that 5,7,4′-trimethoxyflavone, 3,5,7-trimethoxyflavone, 5-hydroxy-3,7,4′-trimethoxyflavone, 5-hydroxy-3,7,3′,4′-tetramethoxyflavone, and 3,5,7,3′,4′-pentamethoxyflavone specifically target SRC, AKT1, and PI3KR1 with binding energy scores detailed in Table 2. Furthermore, Appendix A provide representative three- and two-dimensional structures of selected complexes, respectively. Hydrogen bonds were observed between bioactive compounds and SRC, specifically involving the amino acid residues LYS39 and MET85 (Figure 4). Similarly, all bioactive compounds from KP formed hydrogen bonds with AKT1 via LYS267, except 5,7,4′-trimethoxyflavone, which interacted with both LYS267 and ASN52. Additionally, VAL850 played a key role in hydrogen bonding between all KP bioactive compounds and PIK3R1. Notably, 5-hydroxy-3,7,3′,4′-tetramethoxyflavone and 3,5,7,3′,4′-pentamethoxyflavone formed additional hydrogen bonds with PIK3R1 at LYS801, VAL850, and ASP932, further emphasizing their potential role in modulating PI3K-AKT-related signaling pathways. These findings suggest that these compounds have modulated binding affinities to key signaling proteins, further supporting their potential role in modulating inflammatory pathways in psoriasis.

### 2.5. Anti-Proliferation Effects of Bioactive Compounds in KP on HaCaT and RAW264.7 Cells

The sulforhodamine B (SRB) assay was used to evaluate the effects of five compounds on HaCaT and RAW264.7 under inflammatory conditions induced by LPS (1 μg/mL). LPS treatment significantly increased cell proliferation in both cell types. As shown in Figure 5A, 5,7,4′-trimethoxyflavone and 3,5,7-trimethoxyflavone significantly decreased HaCaT cell proliferation in a dose-dependent manner in all concentrations (0, 5, 10, 20, and 40 μg/mL) in the presence of LPS. 5-hydroxy-3,7,4′-trimethoxyflavone showed minimal effects, while 5-hydroxy-3,7,3′,4′-tetramethoxyflavone and 3,5,7,3′,4′-pentamethoxyflavone had a trend toward decreased cell proliferation without any statistical significance. In addition, 5,7,4′-trimethoxyflavone and 3,5,7-trimethoxyflavone significantly reduced cell proliferation in RAW264.7 cells. 5-hydroxy-3,7,3′,4′-tetramethoxyflavone and 3,5,7,3′,4′-pentamethoxyflavone also decreased cell proliferation only at 5 μg/mL, whereas 5-hydroxy-3,7,4′-trimethoxyflavone had a trend to decrease cell proliferation but without statistical significance (Figure 5B). These findings highlight 5,7,4′-trimethoxyflavone and 3,5,7-trimethoxyflavone as the most bioactive, inhibiting cell proliferation in both HaCaT keratinocytes and RAW264.7 macrophages under inflammatory conditions.

### 2.6. Effects of Bioactive Compounds in KP on RAW264.7 Cell Nitric Oxide Production and HaCaT Cell Migration

To investigate the potential anti-inflammatory effects of 5,7,4′-trimethoxyflavone and 3,5,7-trimethoxyflavone, RAW264.7 macrophages were stimulated with LPS (1 μg/mL) in the presence of increasing concentrations (5, 10, 20, and 40 μg/mL) of each compound, and nitric oxide (NO) levels were measured (Figure 6). LPS stimulation significantly increased NO production compared to the control group. Treatment with 5,7,4′-trimethoxyflavone significantly reduced NO levels in a dose-dependent manner, with a significant decrease observed at 40 μg/mL, indicating its potential anti-inflammatory activity. In contrast, 3,5,7-trimethoxyflavone did not significantly affect NO production at any tested concentration. The anti-migration properties of the two methoxyflavones were evaluated using a scratch assay in HaCaT cells. In Figure 7A,B, microscopic images showed a visible reduction in cell migration in both treatment groups, particularly at higher concentrations. 5,7,4′-trimethoxyflavone significantly decelerated wound closure at 12 and 24 h compared to the initial time point. Similarly, 3,5,7-trimethoxyflavone also inhibited wound healing, with a statistically significant increase in closure observed at 24 h (Figure 7C). These results suggest that 5,7,4′-trimethoxyflavone exhibits both anti-inflammatory and wound-healing effects, while 3,5,7-trimethoxyflavone primarily inhibits wound healing without significantly modulating NO production. Further studies are needed to elucidate the molecular mechanisms underlying these effects.

### 2.7. Effect of Bioactive Compounds in KP on Protein Expressions of Chemokine Signaling Pathways

To investigate the effect of 5,7,4′-trimethoxyflavone and 3,5,7-trimethoxyflavone on key signaling pathways involved in cell survival, proliferation, and inflammation, including AKT, ERK, mitogen-activated protein kinase kinase (MEK), and SRC in HaCaT in the inflammatory condition, the cells were treated with LPS followed by each compound. As shown in Figure 8A–C, LPS stimulation enhanced the phosphorylation of AKT, ERK, MEK, and SRC, indicating activation of these pathways in HaCaT cells. Although neither compound induced drastic changes, 5,7,4′-trimethoxyflavone and 3,5,7-trimethoxyflavone had a negligible decrease effect on the phosphorylation of AKT and SRC at a high dose, respectively. These data suggest that both compounds weakly suppress and regulate the effect on SRC-related pathways, which may play a role in their wound-healing properties.

## 3. Discussion

In this study, we employed an integrated network pharmacology approach combined with molecular docking to investigate the anti-psoriatic effects of methoxyflavones from *Kaempferia parviflora* (KP). Our findings reveal that these compounds interact with key target proteins, influencing the chemokine signaling pathway and PI3K-AKT signaling pathways, thereby identifying them as potential agents in inflammation-related pathways. Experimental validation demonstrated that methoxyflavones, particularly 5,7,4′-trimethoxyflavone and 3,5,7-trimethoxyflavone, effectively inhibited the proliferation and migration of HaCaT keratinocytes and RAW 264.7 macrophages, both critical in psoriasis pathogenesis. Although the effect of a single methoxyflavone on AKT, ERK, MEK, and SRC signaling was moderate, the possible additive or synergistic interactions among methoxyflavones in modulating these pathways warrant further investigation.

Psoriasis is a chronic inflammatory skin disease driven by dysregulated chemokine signaling and immune cell recruitment, leading to keratinocyte hyperproliferation and sustained inflammation. Targeting key signaling pathways involved in psoriasis pathogenesis is a promising strategy for developing effective multi-target therapeutics [32,33]. In this study, we employed an integrated network pharmacology approach combined with molecular docking to investigate the anti-psoriatic potential of methoxyflavones in KP. Experimental validation further confirmed their ability to modulate inflammation, inhibit cell proliferation, and suppress keratinocyte migration under inflammatory conditions.

In the present study, we identified potential bioactive compounds of KP through literature reviews [27,28,30] with the traditional Chinese medicine systems pharmacology database and analysis platform (TCMSP) [34] to investigate the chemokine signaling pathway in the treatment of psoriasis. Based on this information, we systematically screened the components and targets of KP, identifying 16 active components (Table 1 and Appendix A). Corresponding targets were selected using Lipinski’s rules. Toxicity predictions showed no carcinogenicity, hepatotoxicity, or acute oral toxicity, except for 5,7-dimethoxyflavone, which exhibited moderate mutagenicity. Concurrently, we screened a total of 232 potential target genes obtained for 16 identified constituents of KP, revealing 64 overlapping genes linked to psoriasis pathogenesis, as determined using SwissTargetPrediction, STITCH, and TargetNet. Network analysis conducted with Cytoscape highlighted 5,7,4′-trimethoxyflavone as a key bioactive compound that interacts with chemokine signaling and the PI3K-AKT pathways.

The analysis of 64 overlapping genes between 232 KP targets and 1084 psoriasis-related chemokine genes, as identified using a Venn diagram, led to the construction of a PPI network based on these target genes (Figure 2). Gene Ontology (GO) functional analysis and KEGG pathway enrichment analysis revealed that these target genes are involved in several biological processes, mainly regulating inflammatory responses, protein phosphorylation, peptidyl-tyrosine phosphorylation, and protein autophosphorylation. In addition, the molecular function is also involved in transmembrane receptor protein tyrosine kinase activity, protein tyrosine kinase activity, protein kinase activity, protein homodimerization activity, and enzyme binding. The cellular components and molecular functions exhibit a connection to the plasma membrane and transmembrane receptor protein tyrosine kinase activity, respectively. This association is possibly linked to the mechanism of chemokine signaling through autophosphorylation and its role in inflammation-induced autophosphorylation. The KEGG enrichment analysis revealed several signaling pathways that play a role in the development and progression of psoriasis, including the chemokine signaling pathway and the PI3K-AKT signaling pathways, highlighting their critical modulatory roles in the pathogenesis of the disease.

The pathogenesis of psoriasis is profoundly affected by the interaction between immune cells and keratinocytes, as evidenced by the engagement of cytokines and their receptors [35]. The recruitment and activation of immune cells at sites of inflammation rely on chemokines secreted by keratinocyte effector cells. These chemokines operate by initiating downstream signaling pathways, such as JAK/STAT, SRC family activation, and PI3K-AKT [36]. Our study identified the top 10 genes with the highest degree values (*TNF*, *AKT1*, *SRC*, *CASP3*, *ESR1*, *MMP9*, *PPARG*, *PIK3R1*, *CDC42*, and *RELA*) as core targets of KP, linking them to psoriasis pathogenesis. Several studies have revealed that SRC family activation and PI3K/AKT signaling promote epidermal hyperplasia and play distinct regulatory roles in psoriasis immune cells [37,38]. In addition, our study also highlighted the significance of target genes implicated in signaling downstream of the SRC/PI3K/AKT pathways, including glycogen synthase kinase-3 beta (GSK-3β), inhibitor of nuclear factor kappa B kinase subunit beta (IKBKB), and NF-κB. These genes are involved in cytokine production, cellular growth, differentiation, migration, cell survival, and apoptosis [39]. The predicted targets of KP compounds are associated with focal adhesion kinase (FAK), cell division control protein 42 homolog (CDC42), and Ras-related C3 botulinum toxin substrate 1 (RAC1), which are activated through SRC and PI3K signaling pathways that regulate the cytoskeleton of the cell [40]. Thus, the identification and development of the key effective methoxyflavones from KP that could inhibit this signaling pathway and its downstream signaling may help improve treatment efficacy for psoriasis.

Among the predicted compounds, 5,7,4′-trimethoxyflavone, which is one of the polymethoxyflavones found in KP, exhibits strong anti-inflammatory properties [41,42]. It protects human dermal fibroblasts (HDFs) from TNF-α-induced damage [43], inhibits LPS-induced RAW264.7 cells [29], and suppresses IL-1β-induced inflamed human knee-derived chondrocytes [44]. Additionally, 3,5,7-trimethoxyflavone reduced oxidative stress and pro-inflammatory cytokines in HDFs [45], while 5-hydroxy-3,7,4′-trimethoxyflavone also demonstrates anti-inflammatory activity [29,46]. Other KP-derived polymethoxyflavones have been shown to modulate inflammation and support skin health, such as 5,7,4′-trimethoxyflavone, 5-hydroxy-3,7,3′,4′-tetramethoxyflavone, and 3,5,7,3′,4′-pentamethoxyflavone [43,47,48]. Furthermore, 5,7,4′-trimethoxyflavone and 3,5,7,3′,4′-pentamethoxyflavone isolated from *K. parviflora* rhizomes inhibit IL-6, IL-8, and matrix metalloproteinase-1 (MMP-1) secretion in a human ex vivo skin model [49]. However, evidence is still lacking regarding the involvement of the key target KP in chemokine-PI3K-AKT signaling-related psoriasis. Therefore, further research is necessary to confirm the findings from computational simulation studies and in vitro studies.

The flavone backbone in methoxyflavones of KP consists of a 15-carbon structure with A- and B-rings (benzene) interconnected by a pyrone C-ring. The standard atom numbering sites predominantly assign positions to methoxy (–OCH_3_) and hydroxy (–OH) groups primarily, as shown in Appendix A. For example, 3,5,7-trimethoxyflavone derivatives and 5,7,4′-trimethoxyflavone exemplify the A-ring C3/C5/C7 and B-ring C3′/C4′ configurations. These substitution patterns, critical for bioactivity (e.g., anti-inflammatory and anticancer effects), correspond with NMR and GC-MS data from previous studies [30,50]. The structure of 5,7,4′-trimethoxyflavone improves molecular planarity and promotes protein interactions pertinent to antitumor mechanisms, including programmed death-ligand 1 (PD-L1) degradation [51] and caspase-3 activation [52]. In contrast, the C3 methoxy group in 3,5,7-trimethoxyflavone reduces planarity but confers specificity for inhibiting TNF-α-induced MMP-1 secretion in dermal fibroblasts, supporting its dermatoprotective effects with low cytotoxicity [43]. These findings highlight how methoxy substitution patterns modulate the pharmacological profiles of KP-derived flavones. Consequently, we performed a molecular docking analysis to investigate these bioactive compounds and their corresponding targets.

This study employed molecular docking simulations to investigate the binding interactions of principal compounds with associated proteins, based on the premise that the chemokine signaling pathway and the PI3K-AKT signaling pathways are pivotal targets in psoriasis treatment. The predictions indicate that 5,7,4′-trimethoxyflavone, 3,5,7-trimethoxyflavone, 5-hydroxy-3,7,4′-trimethoxyflavone, 5-hydroxy-3,7,3′,4′-tetramethoxyflavone, and 3,5,7,3′,4′-pentamethoxyflavone possess the highest number of target genes and are associated with core chemokine signaling pathways (e.g., SRC-PI3K-AKT signaling pathways). All the binding energies of these compounds were less than −5.00 kcal/mol, indicating potential binding (Table 2). The interaction points suggest that these compounds may be significant in the primary treatment of psoriasis. Our molecular docking study reveals distinct interaction patterns that highlight structure–activity relationships, with hydroxylated methoxyflavones having greater binding affinities. Specifically, 5-hydroxy-3,7,3′,4′-tetramethoxyflavone binds to SRC with a binding affinity of −6.54 kcal/mol and forms hydrogen bonds with LYS39 and MET85. The enhanced ligand–protein interaction may result from the hydroxyl group at position C5 establishing additional hydrogen bonds. 3,5,7,3′,4′-pentamethoxyflavone exhibits optimal binding to PIK3R1 at −6.66 kcal/mol, establishing hydrogen bonds with LYS801, VAL850, and ASP932. Numerous methoxy alterations, particularly at the B-ring positions 3′ and 4′, may enhance spatial orientation within the PIK3R1 binding pocket (Appendix A). Our two primary chemicals exhibit distinct binding affinities. 5,7,4′-trimethoxyflavone has superior binding affinity to PIK3R1 (−6.11 kcal/mol) compared to 3,5,7-trimethoxyflavone (−5.82 kcal/mol), while 3,5,7-trimethoxyflavone demonstrates enhanced binding to AKT1. The differential binding patterns correspond with our experimental results on pathway-specific effects. The augmentation of 5,7,4′-trimethoxyflavone binding to PIK3R1 may elucidate its influence on AKT phosphorylation, whereas its preferred affinity for AKT1 suggests a direct contact mechanism. Hydrogen bonding with significant residues such as LYS267 in AKT1 and VAL850 in PIK3R1 across several compounds suggests methoxyflavone binding sites. The molecular docking findings endorse the advancement of methoxyflavone derivatives as specific modulators of inflammatory and proliferative signaling pathways.

In the experimental validation, 5 of 16 compounds from KP were selected for effect validation based on information from ADMETlab 2.0, confirming their toxicological profile with no indications of carcinogenicity, hepatotoxicity, or acute oral toxicity. Additionally, we evaluated their oral bioavailability (OB%) and drug-likeness (DL) properties (Appendix A), further supporting their potential as drug-like candidates for therapeutic applications. The study demonstrated that 5,7,4′-trimethoxyflavone and 3,5,7-trimethoxyflavone significantly inhibit the proliferation of both HaCaT cells and RAW 264.7 macrophages, indicating their potential as anti-inflammatory and anti-proliferative agents for psoriasis treatment. These compounds also suppress HaCaT cell migration, a critical factor in epidermal hyperplasia, thereby supporting their therapeutic relevance. Furthermore, 5,7,4′-trimethoxyflavone reduces nitric oxide (NO) production in LPS-stimulated macrophages (Figure 5), consistent with previous studies showing its anti-inflammatory and immune-modulating effects [30,53]. In addition, 3,5,7-trimethoxyflavone also suppressed NO production in RAW 264.7 cells, indicating its potential role in modulating inflammation [54]. However, the other compounds showed a trend toward reducing NO production in RAW 264.7 cells (Appendix A). Their ability to target multiple inflammatory processes positions them as promising candidates for further investigation in psoriasis therapy.

From the analysis of cell proliferation and NO production, two compounds were selected for wound healing and protein expression analysis. The wound-healing assay demonstrated that 5,7,4′-trimethoxyflavone and 3,5,7-trimethoxyflavone significantly inhibited HaCaT cell migration (Figure 7). These data suggest their potential to suppress epidermal hyperplasia in psoriasis. This effect may be mediated through chemokine signaling, PI3K-AKT pathways, and SRC family kinases, which are involved in regulating cell motility and tissue remodeling [5,6]. Further, in vivo studies are needed to validate these effects in psoriasis models [55]. Molecular docking studies demonstrated strong binding affinities of methoxyflavones to key signaling proteins, including SRC, AKT1, and PIK3R1, suggesting structural compatibility with their catalytic or regulatory domains. However, Western blot analysis revealed only modest reductions in phosphorylation of AKT, ERK, MEK, and SRC following LPS stimulation (Figure 8), indicating limited functional inhibition under inflammatory conditions. This discrepancy highlights a critical mechanistic ambiguity. While computational models predict direct kinase interactions, the observed anti-inflammatory effects may alternatively arise from upstream receptor modulation or compensatory pathway crosstalk. This finding suggests that while these compounds show promising interactions with their targets, their efficacy may be improved through further combination strategies. Moreover, the observed multi-target effects could result from additive or synergistic mechanisms rather than solely from inhibiting a single pathway. This perspective aligns with the network pharmacology approach, which suggests that phytochemicals exert their effects by modulating multiple targets simultaneously rather than acting on a single molecular entity. Further studies are needed to investigate whether methoxyflavones modulate upstream signaling pathways, such as NF-κB, STAT3, or mitogen-activated protein kinases (MAPKs), which are well-established drivers of inflammation and immune activation in psoriasis [9].

While our study provides evidence supporting the anti-psoriatic potential of methoxyflavones, some limitations should be acknowledged. First, the in vitro nature of our experiments does not fully represent the complex immune microenvironment of psoriasis lesions, which points to the importance of in vivo studies. Additionally, while molecular docking and network pharmacology offer helpful details about potential target interactions, experimental validation of these mechanisms at the protein level remains necessary. Future studies should focus on evaluating methoxyflavones in psoriasis animal models, exploring synergistic effects between multiple compounds, and conducting high-throughput screening of additional KP-derived flavones for enhanced therapeutic efficacy.

## 4. Materials and Methods

### 4.1. Screening the Bioactive Compounds and Target Prediction of KP

Information on the methoxyflavones was acquired from previous research related to KP constituent identification [24,27,28,29,30,56]. Data on the chemical structure of bioactive compounds, including molecular structures, canonical SMILES, and their SDF files, were retrieved from the PubChem and ZINC databases. Subsequently, the drug-likeness properties (Lipinski’s rule) and oral bioavailability score ≥ 0.50 were used to screen identified compounds by the Swiss ADME program [34]. Furthermore, the toxicity characteristics of the main bioactive compounds in KP were investigated using an online pharmacokinetics and toxicity prediction program available online (ADMETlab 2.0). To predict the potential therapeutic targets of corresponding compounds, SwissTargetPrediction, with a probability value ≥ 0.1 and a confidence score ≥ 0.15 in the STITCH database, and TargetNet database, with the species limited to *Homo sapiens*, was applied in this model. Related target gene information, including gene name, gene ID, and gene symbol belonging to *Homo sapiens*, was from UniProtKB.

### 4.2. Screening for Psoriasis-Related Chemokine and Potential Target Genes for KP

The genes of targets associated with “psoriasis” and “chemokine” were retrieved from the following three databases: the NCBI Gene database, the GeneCards database (https://www.genecards.org/, accessed on 31 July 2024), and the Kyoto Encyclopedia of Genes and Genomes (KEGG) database. All duplicated genes were removed from the final gene list, and UniProtKB was employed to obtain the standard name of the target gene, with the organism specified as ‘*Homo sapiens*’. Then, the predicted target genes of KP compounds and psoriasis-related chemokine signaling pathway targets were intersected. A Venn diagram was constructed to extract the common target genes for subsequent analysis using the Venny tool [31].

### 4.3. GO Enrichment and KEGG Pathway Analysis

To further understand the function of core target genes and the main signaling pathways, the Database for Annotation, Visualization, and Integrated Discovery (DAVID) database 6.8 was used to collect Gene Ontology (GO), including cellular components, biological processes, and molecular function [57], as well as KEGG pathway enrichment analyses [58,59]. The filtering thresholds for the retrieved results were set at *p* < 0.05.

### 4.4. Screening of Core Targets and Protein–Protein Interaction Network Construction

To investigate the protein–protein interaction (PPI) between the bioactive compounds of KP and PCSP gene targets, a functional protein-associated network was conducted using the STRING database version 11.5 (https://string-db.org/, accessed on 31 October 2024) [60]. The protein interaction confidence score threshold was set to 0.4 (medium confidence), with the species limited to ‘*Homo sapiens*’ and disconnected nodes excluded from the network representation. The PPI information was exported in the TSV format and then visualized using Cytoscape 3.10.1 software [61].

### 4.5. Network Construction

All the networks were visualized by Cytoscape 3.10.1, which was an open-source graphical user interface for complex network construction, visualization, and analysis [61]. Nodes represented the active compounds and target genes in the network, while edges indicated the interactions between the active bioactive compounds and their target genes. To elucidate the importance of each compound or target in the network diagram, topological properties, including degree, closeness, and betweenness, were calculated using the network analyzer plugin.

### 4.6. Molecular Docking

The three-dimensional structures of bioactive compounds were downloaded from the NCBI PubChem database in .sdf format (https://pubchem.ncbi.nlm.nih.gov, accessed on 1 November 2024). Each ligand underwent geometric optimization using the Gaussian09w program (Revision A.02) [62], applying with DFT-B3LYP method at 6-311G (d, p) levels [63]. Hydrogen atoms were assigned, and aromatic carbons and rotatable bonds of each ligand were determined through AutoDockTools 1.5.6. The crystal structures of SRC (PDB: 2BDJ), PIK3R1 (PDB: 5Fi4), and AKT1 (PDB: 3O96) were retrieved from the Protein Data Bank (RCSB database). For structures from PDB ID 2BDJ and 3O96, the ligand in complex with the protein in the crystal structure was removed prior to experiments in the next step. To refine receptor structures, loop optimization was performed using the MODELLER webserver (University of San Francisco, San Francisco, CA, USA) within UCSF Chimera-1.14 (RBVI, UCSF, San Francisco, CA, USA). The best model was selected using the lowest DOPE-HR score criterion. Before conducting the molecular docking study, the protonation state of the receptor was adjusted by PDB2PQR to provide a structure of amino acid at pH 7.4 [64]. AutoDock 4.2 was used to perform the molecular docking of candidate ligands to promising targets [65]. The docked compound with the lowest AutoDock score in the most populated cluster was further investigated. The binding modes were visualized using Discovery Studio Visualizer version 21.1.0.20298 [66] and PyMOL version 2.5.4 [67].

### 4.7. Chemicals and Reagents

3,5,7-trimethoxyflavone was procured from BOC Sciences (Shirley, NY, USA). 5-hydroxy-3,7,4′-trimethoxyflavone and 3,5,7,3′,4′-pentamethoxyflavone were acquired from Wuhan ChemFaces Biochemical Co., Ltd. (Wuhan, China). Dulbecco’s Modified Eagle Medium (DMEM), fetal bovine serum (FBS), phosphate-buffered saline (PBS), and trypsin-EDTA solution were purchased from Gibco (Grand Island, NY, USA). 5,7,4′-trimethoxyflavone, 5-hydroxy-3,7,3′,4′-tetramethoxyflavone, dimethyl sulfoxide (DMSO), lipopolysaccharide (LPS) from *Escherichia coli*, and sulforhodamine B (SRB) were purchased from Sigma Chemical, Inc. (St Louis, MO, USA). The SuperSignal West Pico Chemiluminescent Substrate was obtained from Invitrogen (Thermo Fisher Scientific Inc., Waltham, MA, USA). Primary antibodies against Phospho-Src (Tyr416) (2101S), Src (2108S), Phospho-Akt (Ser473) (9271S), Akt (9272S), Phospho-MEK1/2 (Ser217/221) (9121S), MEK1/2 (9122S), Phospho-p44/42 MAPK (Erk1/2) (Thr202/Tyr204) (9101S), and p44/42 MAPK (Erk1/2) (9102S) were from Cell Signaling Technology (Beverly, MA, USA). Actin (ab8227), peroxidase-labeled secondary antibodies, anti-rabbit IgG (ab97051), and anti-mouse IgG (ab97046) were purchased from Abcam (Cambridge, U.K.). Protease and phosphatase inhibitors were obtained from Roche Diagnostics, Mannheim, Germany.

### 4.8. Cell Culture

The macrophage cells used in this study were RAW 264.7 obtained from ATCC (Manassas, VA, USA), and the HaCaT cells were purchased from CLS Cell Lines Service GmbH (Eppelheim, Germany). The cells were cultured in DMEM supplemented with 10% fetal bovine serum, penicillin (100 U/mL), and streptomycin (100 μg/mL) at 37 °C under 5% CO_2_. For the experiments, untreated cells served as the control group, while LPS (1 ng/mL) was used to induce inflammation. Each test compound was co-treated with LPS to assess its potential effects under inflammatory conditions.

### 4.9. Cell Proliferation

RAW 264.7 or HaCaT cells were seeded in a 96-well culture plate in 80% confluence and co-treated with LPS and varying concentrations of each compound (5, 10, 20, and 40 μg/mL). After 48 h of incubation, cell proliferation for both cell types was assessed using the SRB assay. Following the treatments, the cells were fixed with TCA and then incubated at 4 °C for one hour. Subsequently, the plates underwent four washes with tap water and were then air-dried. Each well was added 100 μL of 0.057% (*w*/*v*) SRB solution and then incubated at room temperature for 30 min. After this, the plates were washed four times with 1% (*v*/*v*) acetic acid to remove any unbound dye. After the plate was dried, 200 μL of a 10 mM Tris-based solution (pH 10.5) was added to each well to dissolve the dye. The absorbance at 510 nm was measured using a microplate reader (BioTek, Winooski, VT, USA). Compounds were selected for further study based on concentrations below their IC_20_ values [68].

### 4.10. Cell Migration Assay

Cell migration was evaluated using a wound closure assay following the previously described [69]. In brief, HaCaT (2 × 10^6^) cells were seeded in 6-well plates, cultured to 100% confluence, scraped off using a 200 μL pipette tip, and washed out gently with serum-free medium. Cells were treated for each condition. The percentage of the cell closure area was determined at 0, 12, and 24 h under a Nikon Eclipse Ts2 phase-contrast microscope (Nikon Corporation, Tokyo, Japan). ImageJ software, version 1.54, was used to determine the migration distance.

### 4.11. Western Blotting

The treated cells were lysed with a RIPA buffer and measured for protein concentration using a Bradford assay kit, as described in a previous study [27]. To determine protein expression, protein samples were run on SDS-PAGE and then transferred onto nitrocellulose membranes. After that, 5% skim milk diluted in 0.1% Tween-TBS was used to block non-specific binding on the membrane for 1 h at room temperature. The membrane was incubated with the individual-specific primary antibodies at 4 °C overnight. The membrane was washed and incubated with peroxidase-labeled secondary antibodies for 2 h at room temperature and then incubated with the SuperSignal West Pico Chemiluminescent Substrate. The intensity of each band was measured using ImageJ software, version 1.54 (National Institutes of Health, Bethesda, MD, USA).

### 4.12. Statistical Analysis

Statistical analysis was conducted using GraphPad Prism 8.0 (San Diego, CA, USA). The mean ± standard deviation (S.D.) was employed in expressing the data. ANOVA with the post hoc Tukey’s test was used to examine group differences. Three replications (*n* = 3) were performed for each experiment. Statistical significance was defined as follows: * *p* < 0.05, ** *p* < 0.01, *** *p* < 0.001, and **** *p* < 0.0001 relative to either the control group or the LPS-treated group, depending on the experimental conditions.

## 5. Conclusions

In conclusion, our study highlights methoxyflavones as promising candidates for multi-target psoriasis therapy, leveraging their ability to modulate chemokine signaling, suppress keratinocyte hyperproliferation, and reduce inflammation. By integrating computational predictions with experimental validation, we provide a robust theoretical framework for future in vivo, synergistic combinations of methoxyflavones and clinical investigations, supporting the potential application of methoxyflavones as novel, plant-derived therapeutics for inflammatory skin diseases.

## Figures and Tables

**Figure 1 ijms-26-05243-f001:**
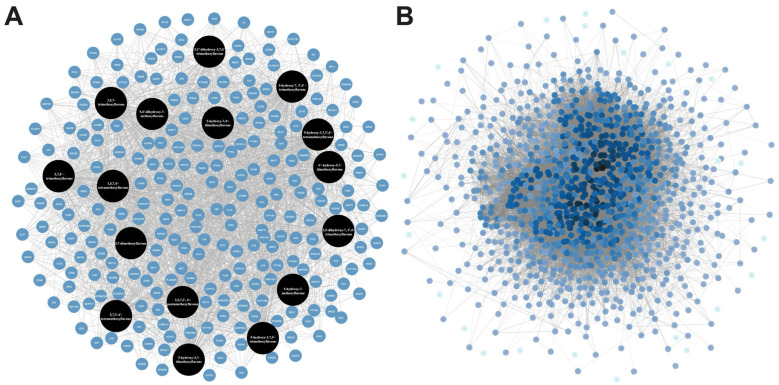
(**A**) The compound–target–disease network constructed using psoriasis-related chemokine target genes of bioactive compounds in KP. (**B**) Protein–protein interaction (PPI) network constructed using psoriasis-related chemokine target genes of bioactive compounds in KP.

**Figure 2 ijms-26-05243-f002:**
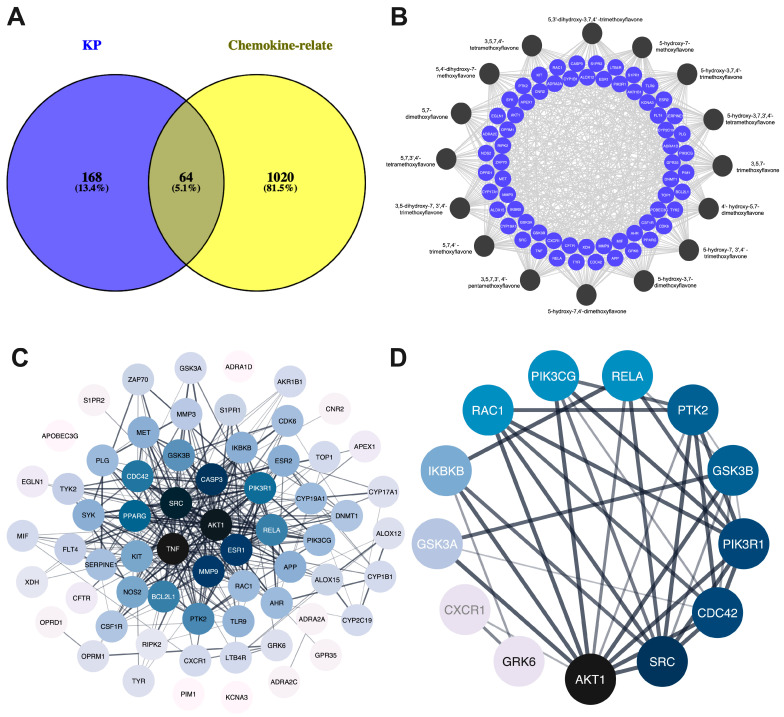
Compound targets the network and protein–protein interaction network. (**A**) Venn diagram showing the intersection targets of KP targets and psoriasis-related chemokine genes. (**B**) Compound targets network. The circular black node represents the KP bioactive compounds, and light blue represents the target genes. (**C**) Core target genes of chemokine signaling pathways. (**D**) Black nodes represent the higher degree. Color saturation of the edges represents the confidence score of a functional association.

**Figure 3 ijms-26-05243-f003:**
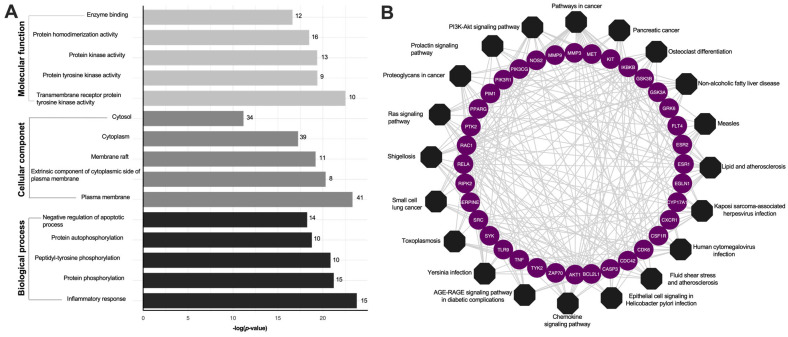
Enrichment analysis of the KP targets. (**A**) Bar plot GO enrichment analysis. The top five significantly enriched GO (−log10 (*p*-value)) terms of the target genes in the molecular function (light gray), cellular component (dark gray), and biological processes (black). (**B**) KP compounds target pathway networks. The circular purple nodes represent the related targets, and the black nodes represent the top twenty pathways.

**Figure 4 ijms-26-05243-f004:**
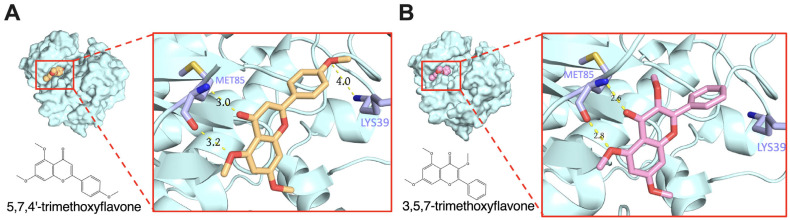
Molecular docking of molecule structures and binding sites between SRC and selected compounds: (**A**) 5,7,4′-trimethoxyflavone and (**B**) 3,5,7-trimethoxyflavone.

**Figure 5 ijms-26-05243-f005:**
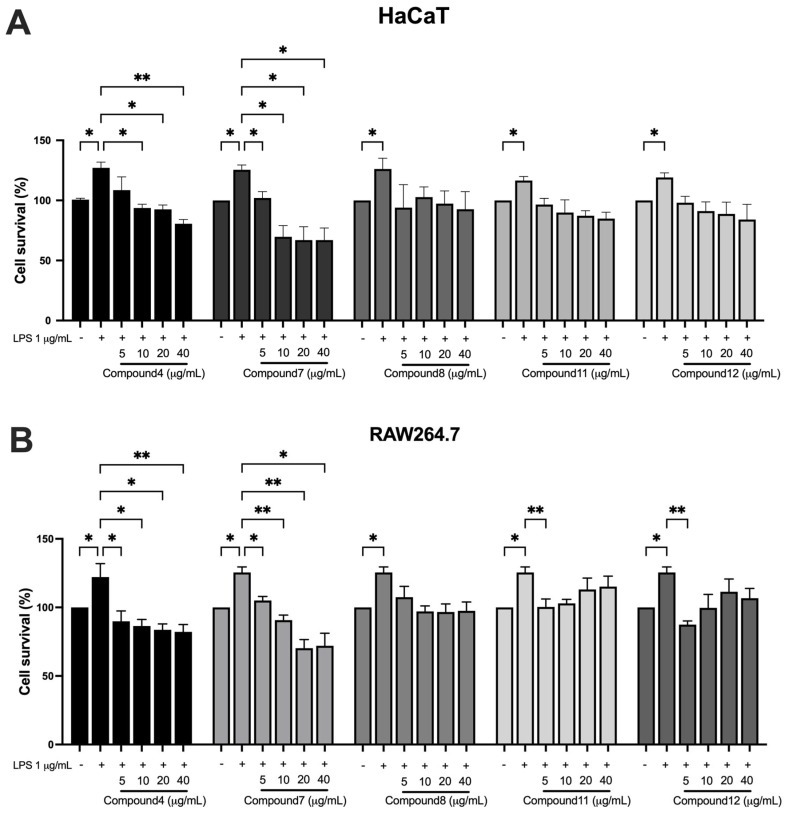
Effects of bioactive compounds in KP on cell proliferation in HaCaT and RAW264.7 cells under inflammatory conditions. (**A**) HaCaT keratinocytes and (**B**) RAW264.7 macrophages were treated with LPS (1 μg/mL) to induce an inflammatory response, followed by exposure to different concentrations (0, 5, 10, 20, and 40 μg/mL) of each compound for 48 h. Data are presented as mean ± S.D. from at least three independent experiments. Statistical significance was determined using one-way ANOVA followed by post hoc analysis (* *p* < 0.05 and ** *p* < 0.01). Compound 4: 5,7,4′-trimethoxyflavone; compound 7: 3,5,7-trimethoxyflavone; compound 8: 5-hydroxy-3,7,4′-trimethoxyflavone; compound 11: 5-hydroxy-3,3′,4′,7-tetramethoxyflavone; and compound 12: 3,5,7,3′,4′-pentamethoxyflavone.

**Figure 6 ijms-26-05243-f006:**
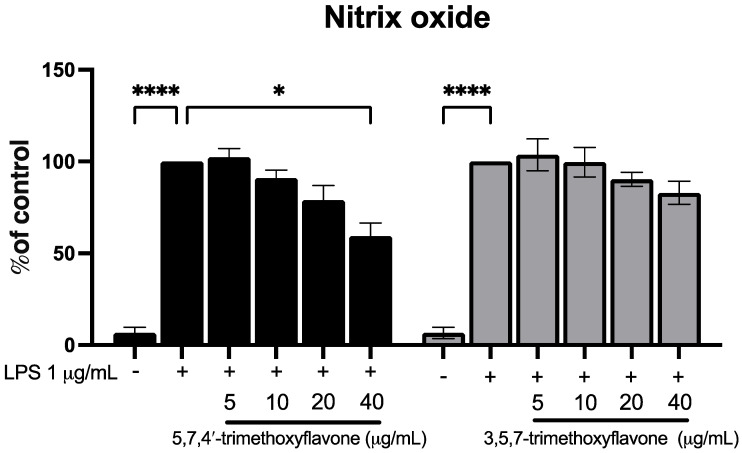
Effects of 5,7,4′-trimethoxyflavone and 3,5,7-trimethoxyflavone on nitric oxide production in LPS-induced RAW264.7 cells. RAW264.7 cells were co-treated with LPS (1 µg/mL) with either 5,7,4′-trimethoxyflavone (left) or 3,5,7-trimethoxyflavone (right) of KP (5, 10, 20, and 40 µg/mL) for 24 h, and nitrite concentrations were measured using the Griess reaction. Data are presented as mean ± S.D. of three independent experiments. Statistical significance was determined using one-way ANOVA followed by post hoc analysis (* *p* < 0.05 and **** *p* < 0.0001).

**Figure 7 ijms-26-05243-f007:**
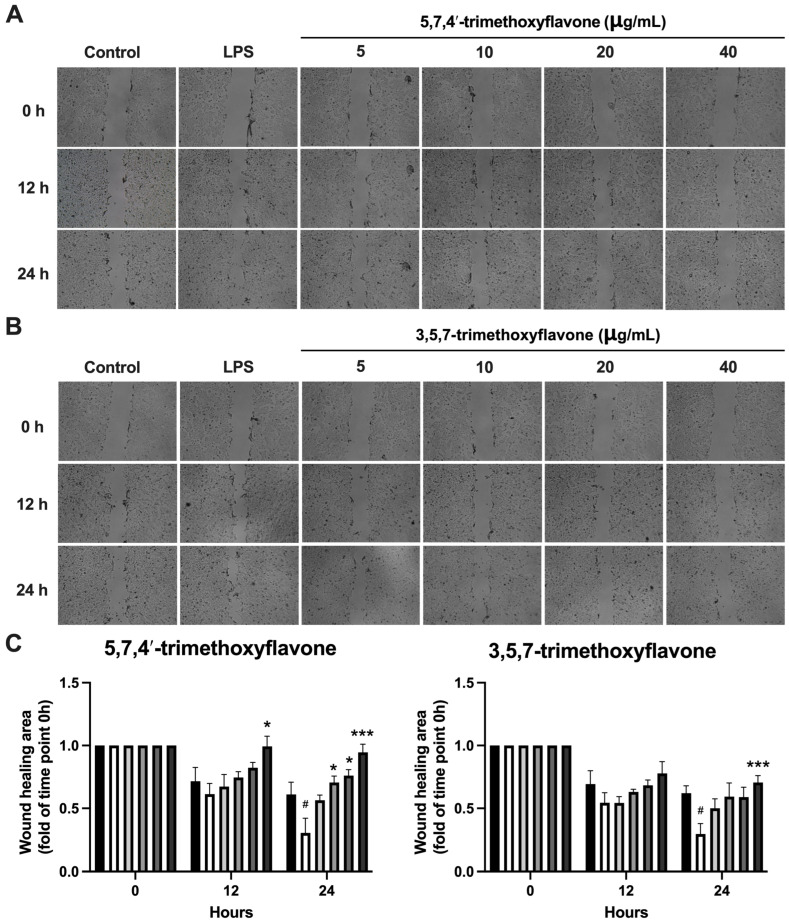
Effects of 5,7,4′-trimethoxyflavone and 3,5,7-trimethoxyflavone on HaCaT Cell Migration. (**A**) Microscopic pictures of HaCaT cell migration assay and (**B**) graphical analysis of the effect of 5,7,4′-trimethoxyflavone and 3,5,7-trimethoxyflavone on keratinocyte migration (the percentage of gap-filled). The cells treated with 15 μg/mL of 5,7,4′-trimethoxyflavone and 3,5,7-trimethoxyflavone were compared with the untreated cells and DMSO-treated cells. (**C**) Quantification of the wound healing area of both compounds. Data are presented as mean ± S.D. of three independent experiments. Statistical significance was determined using one-way ANOVA followed by post hoc analysis (* *p* < 0.05 and *** *p* < 0.001 vs. control and # *p* < 0.05 vs. LPS group).

**Figure 8 ijms-26-05243-f008:**
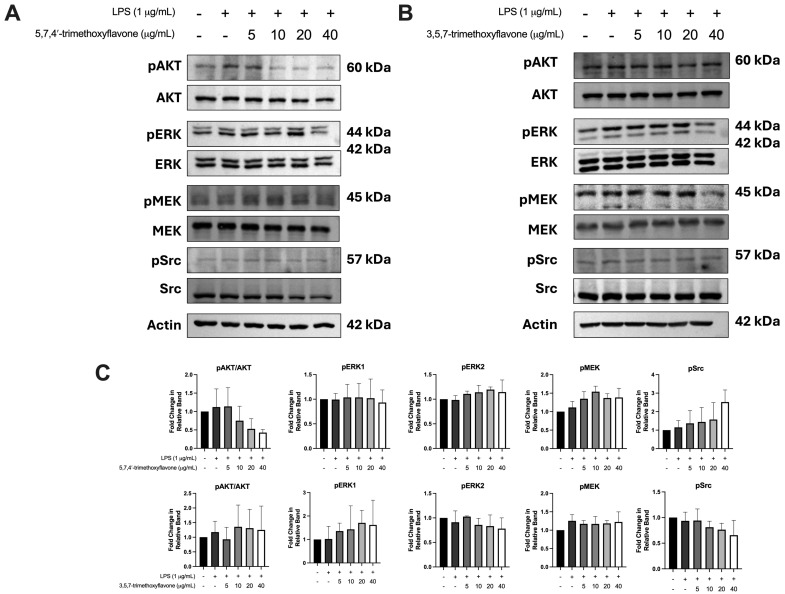
Effects of 5,7,4′-trimethoxyflavone and 3,5,7-trimethoxyflavone on the phosphorylation of key signaling proteins in LPS-induced HaCaT cells. (**A**) Western blot analysis of pAKT, AKT, pERK, ERK, pMEK, MEK, pSRC, and SRC in LPS-stimulated cells treated with different concentrations (5, 10, 20, and 40 μg/mL) of 5,7,4′-trimethoxyflavone. (**B**) Western blot analysis of the same proteins in LPS-stimulated HaCaT cells treated with 3,5,7-trimethoxyflavone at varying concentrations. (**C**) Quantification of the fold change in pAKT/AKT, pERK1, pERK2, pMEK, and pSRC relative to control groups. Data are presented as mean ± S.D. of three independent experiments.

**Table 1 ijms-26-05243-t001:** Information of active ingredients in KP.

No.	Compounds	Molecular Formula	Lipinski Rules				
			MW (<500)	HBA (<10)	HBD (5)	MlogP (<4.15)	TPSA (Å2) (<140)
1	5-hydroxy-7-methoxyflavone	C_16_H_12_O_4_	268.26	4	1	1.33	59.67
2	5,7-dimethoxyflavone	C_17_H_14_O_4_	282.29	4	0	1.57	48.67
3	5-hydroxy-7,4′-dimethoxyflavone	C_17_H_14_O_5_	298.29	5	1	1.01	68.90
4	5,7,4′-trimethoxyflavone	C_18_H_16_O_5_	312.32	5	0	1.25	57.90
5	5,7,3′,4′-tetramethoxyflavone	C_19_H_18_O_6_	342.34	6	0	0.94	67.13
6	5-hydroxy-3,7-dimethoxyflavone	C_17_H_14_O_5_	298.29	5	1	1.01	68.90
7	3,5,7-trimethoxyflavone	C_18_H_16_O_5_	312.32	5	0	1.25	57.90
8	5-hydroxy-3,7,4′-trimethoxyflavone	C_18_H_16_O_6_	328.32	6	1	0.7	78.13
9	3,5,7,4′-tetramethoxyflavone	C_19_H_18_O_6_	342.34	6	0	0.94	67.13
10	5,3′-dihydroxy-3,7,4′-trimethoxyflavone	C_18_H_16_O_7_	344.32	7	2	0.17	98.36
11	5-hydroxy-3,7,3′,4′-tetramethoxyflavone	C_19_H_18_O_7_	358.34	7	1	0.4	87.36
12	3,5,7,3′,4′-pentamethoxyflavone	C_20_H_20_O_7_	372.37	7	0	0.63	76.36
13	3,5-dihydroxy-7,3′,4′-trimethoxyflavone	C_18_H_16_O_7_	344.32	7	2	0.17	98 36
14	5,4′-dihydroxy-7-methoxyflavone	C_16_H_12_O_5_	284.26	5	2	0.77	79.90
15	5-hydroxy-7,3′,4′-trimethoxyflavone	C_18_H_16_O_6_	328.32	6	1	0.7	78.13
16	4′-hydroxy-5,7-dimethoxyflavone	C_17_H_14_O_5_	298.29	5	1	1.01	68.90

MW, molecular weight (g/mol); HBA, Hydrogen Bond Acceptor; HBD, Hydrogen Bond Donor; LogP, Lipophilicity; bioavailability score, the ability of a drug or other substance to be absorbed and used by the body; TPSA (Topological Polar Surface Area).

**Table 2 ijms-26-05243-t002:** The binding energy score of potential target compounds in KP.

Proteins	Compounds	Binding Energy (kcal/mol)	Hydrogen Bonds
SRC (PDB: 2BDJ)	5,7,4′-trimethoxyflavone	−5.71	LYS39, MET85
	3,5,7-trimethoxyflavone	−5.54	MET85
	5-hydroxy-3,7,4′-trimethoxyflavone	−6.03	LYS39, TYR84, MET85
	5-hydroxy-3,7,3′,4′-tetramethoxyflavone	−6.54	LYS39, MET85
	3,5,7,3′,4′-pentamethoxyflavone	−6.40	LYS39, MET85
AKT1 (PDB: 3O96)	5,7,4′-trimethoxyflavone	−5.88	LYS267, ASN52
	3,5,7-trimethoxyflavone	−6.33	LYS267
	5-hydroxy-3,7,4′-trimethoxyflavone	−5.85	LYS267
	5-hydroxy-3,7,3′,4′-tetramethoxyflavone	−6.02	LYS267
	3,5,7,3′,4′-pentamethoxyflavone	−5.83	LYS267
PIK3R1 (PDB: 5Fi4)	5,7,4′-trimethoxyflavone	−6.11	VAL850
	3,5,7-trimethoxyflavone	−5.82	VAL850
	5-hydroxy-3,7,4′-trimethoxyflavone	−6.00	VAL850
	5-hydroxy-3,7,3′,4′-tetramethoxyflavone	−6.33	LYS801, VAL850, SER853, ASP932
	3,5,7,3′,4′-pentamethoxyflavone	−6.66	LYS801, VAL850, ASP932

## Data Availability

Data are contained within the article and Appendix A.

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
