# Peer review of "Network Pharmacology and Molecular Docking-Based Approach to Explore Potential Bioactive Compounds from Kaempferia parviflora on Chemokine Signaling Pathways in the Treatment of Psoriasis Disease"

_ijms, 2025, doi:10.3390/ijms26115243_

Round 1
Reviewer 1 Report
Comments and Suggestions for Authors
In the manuscript, the anti-psoriatic potential of methoxyflavones derived from KP were investigated through an integrated approach combining network pharmacology, molecular docking, and experimental validation. My comments are as follows.
- In line 106, the author mentions “A total of 16 methoxyflavones in KP were obtained through literature searches [27,28,30], and the relevant ADME information is presented in Table 1.” Please provide basis structure and atom numbering sites of KP in the manuscript or Supplementary Data. All subsequent discussions are based on these structures.
- For Molecular docking results, the author only provides the binding energy scores of potential target compounds in KP using a table. I think relevant diagrams (3D or 2D) should be added to further illustrate the relationship between these five methoxyflavonoids and SRC, AKT1, and PI3KR1. In addition, you should choose potential target composites with strong interactions for in-depth discussion.
- In line 494, the author describes “applying with DFT-B3LYP method at 6-311G (d, p) levels B3LYP [60].” Please revise it. The final “B3LYP” seems unnecessary.
Reviewer 2 Report
Comments and Suggestions for Authors
Dear Editor,
Thank you for the opportunity to review this manuscript. I sincerely appreciate the trust you have placed in me. Please find my review comments below.
The submitted manuscript, titled “Network pharmacology and molecular docking-based approach to explore potential bioactive compounds from Kaempferia parviflora on chemokine signaling pathway in the treatment of psoriasis disease," shows potential for publication in this journal. This article presents a well-designed and comprehensive investigation on the anti-inflammatory activity of polymethoxyflavones, specifically 5,7,4-trimethoxyflavone (C1) and 3,5,7-trimethoxyflavone (C2), in LPS-stimulated macrophages. The work aims to elucidate the molecular mechanisms involved, with a focus on MAPK, AKT, and Src pathways. While the study presents promising findings, several areas require clarification and refinement. Below are comments and suggestions for revision as major.
Please address the following comments:
- No clear confirmation of the chemical identity or purity (e.g: HPLC, NMR) of C1 and C2 is presented. This is essential for reproducibility and should be at least mentioned or referenced.
- The Western blot images in Figure 7 show multiple lanes but labeling and alignment across replicates should be clarified. Some bands (e.g: pSrc and Src) appear faint or inconsistent. Please indicate whether blots were repeated and provide quantification for better understanding.
- There is no mention of internal replicates (n-values) or statistical power. Please report the number of biological replicates and clarify how significance was assessed.
- The supplementary data (supplementary.zip) should be better integrated. Several key experiments (e.g: docking, cytotoxicity) are described there but lack reference in the main text.
- The authors conclude that these polymethoxyflavones “inhibit inflammation” mainly via these pathways. Additional inhibitors or rescue experiments would strengthen this claim and It is unclear whether the inhibition is direct or due to upstream receptor interference. Consider addressing this in discussion.

- The manuscript would benefit from language polishing. For instance, phrases like “exerts inflammation regulation” could be replaced with “modulates inflammation.”
- Phrases like “kinds of,” “can do,” or “it was found that” should be replaced with formal scientific expressions.
- Ensure all abbreviations (e.g: LPS, MAPK) are spelled out at first use.
Round 2
Reviewer 1 Report
Comments and Suggestions for Authors
In this version most of my comments were taken into account. I agree to recommend accepting this paper in present form.